

# Diversity, antibacterial and antioxidant activities of fungi associated with *Apis cerana*

Pu Cui[1], Guanxiu Guan[1], Zhuoting Gan[2] and Ting Yao[1]

[1] College of Life and Environmental Sciences, Huangshan University, Huangshan, Anhui, China
[2] School of Tourism, Huangshan University, Huangshan, Anhui, China

## ABSTRACT

Insect-associated fungi are a treasure trove of natural active compounds. Nevertheless, the diversity and biological activities of fungi associated with *Apis cerana* have not been studied in depth. Here, we investigated fungal diversity in the *A. cerana* gut and honeycomb using a combination of culture-dependent and -independent methods. A total of 652 fungal operational taxonomic units belonging to five phyla and 334 genera were detected in the samples. Significant differences were found in the fungal communities of the honeybee gut and honeycomb—the genera *Fusarium*, *Stenocarpella*, and *Botrytis* were dominant in the gut, whereas *Botrytis*, *Periconia*, and *Aspergillus* were dominant in honeycomb. A total of 28 fungal strains were isolated from honeybee gut, head, and honeycomb, belonging to two phyla, four classes, and 10 genera. Most of these isolates were identified as *Aspergillus*, *Penicillium*, and *Cladosporium* spp. The antibacterial and antioxidant activities of crude extracts of their fermentation broths were investigated. Extract from *A. subramanianii* ZFCZ33 exhibited the best antibacterial activities against *Staphylococcus aureus*, *Pseudomonas aeruginosa*, *Escherichia coli*, and *P. syringae* pv. *Actinidiae* with the disc diameter of inhibition zone diameter (IZD) of 24.33, 15.33, 17.00, and 25.33 mm, respectively. Extract from *P. adametzioides* ZFCZ03 had a free radical scavenging rate of 89.71% in assay with 2,2-diphenyl-1-picrylhydrazyl, and that from strain ZFT07 had a free radical scavenging rate of 97.13% in assay with 2,2′-azino-bis(3-ethylbenzothiazoline-6-sulfonic acid). Our results preliminarily elucidate the fungal diversity of *A. cerana* gut and honeycomb and indicate that honeybee-associated fungi have antibacterial and antioxidant activities. This study provides a basis for further development and use of honeybee-associated fungi.

## INTRODUCTION

Honeybees are social insects that play a vital role in pollination (*Potts et al., 2016*; *Eisenhauer, Bonn & Guerra, 2019*). Microbiomes occupy specific niches within the honeybee gut that play a crucial role in maintaining bee health, including in growth and development, immune function, metabolism, and protection against pathogens (*Raymann & Moran, 2018*; *Motta & Moran, 2024*). Much research has focused on honeybee gut bacteria, with gut fungi being relatively ignored. However, several studies suggest that honeybee gut

Corresponding author
Ting Yao, yting@hsu.edu.cn

fungi have a positive effect on food digestion and development, and assist in maturation of royal jelly (*Yun et al., 2018*; *Callegari et al., 2021*; *Rutkowski, Weston & Vannette, 2023*). Some fungi are also found in beehives, pollen, and honey (*Menezes et al., 2015*; *Hsu, Wang & Wu, 2021*; *Tiusanen, Becker-Scarpitta & Wirta, 2024*).

Usually, one or a combination of high-throughput sequencing and cultivation methods is used to study the fungal diversity of honeybees (*Cui et al., 2022a*; *Santos et al., 2023*; *Chow et al., 2024*). Some studies have found the presence of fungi in specific tissues of honeybees, such as the gut, head, and cuticle (*Thamm et al., 2023*; *Tejerina et al., 2023*; *Agarbati et al., 2024*). Honeybees are a source of novel fungal species, for example, *Ascosphaera callicarpa* isolated from larval feces (*Wynns, Jensen & Eilenberg, 2013*); *Arthrinium locuta-pollinis*, *Chrysosporium alvearium*, *Nigrograna locutapollinis*, and *Trichoderma pollinicola* isolated from bee pollen (*Zhao et al., 2018*); and *Strongyloarthrosporum catenulatum*, *Helicoarthrosporum mellicola*, *Oidiodendron mellicola*, *Skoua asexualis*, *Talaromyces basipetosporus*, *T. brunneosporus*, and *T. affinitatimellis* isolated from honey (*Rodríguez-Andrade et al., 2019*). Moreover, metabolites of honeybee-associated fungi exhibit a wide range of biological properties, including antioxidant and antibacterial activities (*Cui et al., 2022a*; *Vocadlova et al., 2023*). Many novel active compounds have been isolated from the metabolites of honeybee-associated fungi (*Elbanna et al., 2021*; *Elbanna, Khalil & Capon, 2021*; *Bulatov et al., 2022*; *Dang et al., 2022*).

Species, environmental conditions, and food sources can cause differences in honeybee gut microbial communities (*Gaggìa et al., 2023*). *Apis cerana* is a widely bred honeybee species for honey production and crop pollination in China (*Chen et al., 2021*). However, to our knowledge, there are few relevant studies about the fungal diversity of *A. cerana* and the bioactivities of these fungi. *A. cerana* is a local species of honeybee that has adapted over a long period in the mountainous areas of southern Anhui province, China. Here, we investigated the fungal diversity of *A. cerana* from southern Anhui province using culture-dependent and -independent methods, and explored the antibacterial and antioxidant activities of cultivable honeybee-associated fungi.

## MATERIALS & METHODS

### Sample collection and preparation

*A. cerana* were collected from one hives of an apiary in Xiuning, Huangshan, China (29°72′N, 117°90′E) in September 2023. A total of 70 honeybee samples were immediately transferred to the laboratory and starved for 24 h, then samples (including honeybee and honeycomb) were stored at −20 °C until DNA extraction and fungal isolation. The honeybee and honeycomb samples were surface disinfected with 75% ethanol for 2 min and then rinsed thrice with sterile water. Subsequently, the guts and heads of honeybees were dissected using sterile forceps.

### High-throughput sequencing

Fungal DNA was respectively extracted from seven honeybee gut samples and one g of honeycomb samples using an E.Z.N.A.® Soil DNA Kit (Omega Bio-tek, Norcross, GA, USA) according to the manufacturer's instructions. Fungal DNA concentration and

quality were detected by 1% agarose gel electrophoresis. The internal transcribed spacer (ITS) region was amplified using primers ITS1F and ITS2R (*Ou et al., 2019*). The PCR reaction system was 20 μL: four μL 5× FastPfu Buffer, two μL 2.5 mM dNTPs, 0.8 μL forward primer (five μM), 0.8 μL reverse primer (five μM), 0.4 μL FastPfu polymerase, and 10 ng template DNA, with the remaining volume filled with double-distilled $H_2O$. PCR reactions were conducted using the program: 4 min of denaturation at 94 °C; 25 cycles of 30 s at 94 °C, 30 s at 55 °C for annealing, and 1 min at 72 °C for elongation; and a final extension for 10 min at 72 °C. Amplicons were extracted from 2% agarose gels and purified using an AxyPrep DNA Gel Extraction Kit (Axygen Biosciences, Union City, CA, USA) according to the manufacturer's instructions. The purified PCR products were quantified and homogenized, and then a PacBio sequencing library was constructed. Amplicon sequencing was performed by Shanghai Biozeron Biotechnology Co. Ltd. (Shanghai, China).

Smrtlink (v 9) was used to preprocess the raw sequencing data. Then operational taxonomic units (OTUs) were clustered with a 98.65% similarity cutoff using UPARSE (v 7.1) and chimeric sequences were identified and removed using UCHIME. Subsequently, the taxonomy of each fungal ITS gene sequence was analyzed using the UNITE database with a confidence threshold of 70%. Finally, the community composition at each classification level, the alpha diversity, and beta diversity of different samples were analyzed and compared. The raw data are available in the NCBI Short Reads Archive (BioProject PRJNA1213968).

## Isolation of cultivable fungi

According to an earlier report (*Cui et al., 2022a*), the plate smearing method was used to isolate fungal strains from the samples. Specifically, the guts and heads of seven sterilized honeybees and 1 g of honeycomb samples were separately ground in 10 mL sterile water to give three suspensions (*i.e.,* guts, heads, and honeycomb). Then, the mixtures were diluted ($10^{-1}$, $10^{-2}$, $10^{-3}$, and $10^{-4}$) and aliquots of 100 μL from each dilution were plated onto three isolation media (potato dextrose agar (PDA), malt extract agar, and Sabouraud dextrose agar) containing 50 mg/L ampicillin and 50 mg/L streptomycin sulfate (*Cui et al., 2022a*). Finally, these agar plates were incubated at 28 °C for 7 days in an MGC-350BP-2 light incubator (Shanghai Yiheng Scientific Instruments Co. Ltd. Shanghai, China). Mycelial growth was observed every day, and promptly selected fungal colonies were transferred to fresh PDA plates for purification. The isolated fungal strains were numbered and preserved on PDA slants at 4 °C and in storage tubes at −80 °C.

## Molecular identification of cultivable fungi

Fungal isolates were identified by molecular biological methods according to a previous report with some modifications (*Kong et al., 2023*). Briefly, each fungal isolate was purified on a PDA plate and cultured at 28 °C for 3–4 days. Then, fungal hypha were picked and placed in a 1.5 mL sterile centrifuge tube containing 20 μL of lysis buffer. The samples were lysed in a water bath at 80 °C for 15 min. After cooling, the centrifuge tube was centrifuged at 10,000 r/min for 10 min, and the supernatant was used as a DNA template. The ITS

region of the fungal ribosomal DNA was amplified with primers ITS 1 and ITS 4. The reaction mixture and amplification conditions were the same as previously described (*Cui et al., 2022a*). PCR products were directly sequenced by TSINGKE Biological Technology Co., Ltd. (Nanjing, China). All sequences received were used for BLAST comparison and species identification using strains in the NCBI database. A phylogenetic tree was constructed by the neighbor-joining method using MEGA software v 5.1.

## Preparation of crude extracts from fungal fermentation broth

Using the same method described in section molecular identification of cultivable fungi, each fungal isolate was transferred to a PDA plate and incubated at 28 °C for 3–4 days. Next, each fungus was transferred to a 250 mL conical flask containing 150 mL of liquid potato dextrose medium, and cultured in a shaker at 180 rpm and 28 °C for 7 days (*Kong et al., 2023*; *Wen et al., 2023*). The fermentation broth was obtained by passing the culture broth through four layers of cotton gauze. Then, the fermentation broth was extracted three times with ethyl acetate (1:1 v:v). Finally, crude extracts of fermentation broth were obtained by concentrating the ethyl acetate phase in a vacuum.

## Antibacterial activity

The antibacterial activities of the crude extracts of culturable fungi were determined by the filter paper dispersion method (*Salinas et al., 2018*; *Ghazi-Yaker et al., 2024*; *Wacira et al., 2024*). Each crude extract of a culturable fungus was dissolved in methanol to 10 mg/mL, and passed through a sterile 0.22 μm pore filter. Test bacterial suspension (100 μL, $10^8$ colony-forming units/mL) was pipetted into 100 mL of Luria-Bertani agar (LBA) medium, mixed evenly, and poured into a sterile plate. The test bacterial species included *Staphylococcus aureus*, *Escherichia coli*, *Pseudomonas aeruginosa*, *P. syringae* pv. *actinidiae*, *Bacillus thuringiensis*, and *B. subtilis*. Then, five μL of methanol solution of crude fungal extract was added to a six mm sterile filter paper that was placed on the LBA plate containing the test bacteria. Methanol and gentamicin sulfate were used as negative and positive controls, respectively. Finally, the plates were incubated at 37 °C for 24 h, and the inhibition zone diameters (IZD; in mm) were measured to evaluate antibacterial activity. The experiment was repeated three times.

## Antioxidant activity
### DPPH radical scavenging activity assay
The 2,2-diphenyl-1-picrylhydrazyl (DPPH) radical scavenging assay was conducted using a previous method with modifications (*Boomi et al., 2020*). Briefly, methanol solution of fungal crude extract (40 μL, one mg/mL) was added to a 96-well plate and mixed with DPPH solution (160 μL, 0.2 mM) in the dark for 10 min at room temperature. The absorbance of each well was recorded at 517 nm by using a microplate reader. Methanol was used as a negative control, and ascorbic acid and quercetin were used as positive controls. The test was repeated three times. DPPH radical scavenging activity was calculated using the equation:

Scavenging rate (%) $= (1 - A_1/A_0) \times 100\%$

where $A_0$ is the absorbance of DPPH plus methanol, and $A_1$ is the absorbance of DPPH plus sample solution.

### ABTS radical scavenging activity assay

2,2′-Azino-bis(3-ethylbenzthiazoline-6-sulfonic acid) (ABTS) radical scavenging assay was conducted according to a previous method with modifications (*Liu et al., 2024*). Briefly, ABTS working solution was prepared by dissolving 38.4 mg of ABTS in 10 mL of pure water and mixing that with 10 mL of potassium persulfate solution (2.45 mM). The solution was stored in the dark at 25 °C for 16 h. Methanol solution of fungal crude extract (40 μL, one mg/mL) was added to a 96-well plate and mixed with 160 μL of the ABTS working solution in the dark for 10 min at room temperature. The absorbance of each well was recorded at 734 nm by using a microplate reader. Methanol was used as a negative control, and ascorbic acid and quercetin were used as positive controls. The test was repeated three times. ABTS radical scavenging activity was calculated using the equation:

$$\text{Scavenging rate (\%)} = (1 - A_1/A_0) \times 100\%$$

where $A_0$ is the absorbance of ABTS plus methanol, and $A_1$ is the absorbance of ABTS plus sample solution.

### Hydroxyl radical scavenging activity assay

The hydroxyl radical scavenging activity assay was conducted using a previous method with modifications (*Tang et al., 2021*). Briefly, methanol solution of fungal crude extract (100 μL, one mg/mL) was added to a 1.5-mL centrifuge tube and mixed with $FeSO_4$ solution (100 μL, 6.0 mM), salicylic acid–methanol solution (100 μL, 6.0 mM), and 20 μL of 0.3% H2O2. The resulting solution was filled to one mL with water and then warmed at 35 °C for 30 min in a water bath. The absorbance of the mixture was measured at 510 nm. Methanol was used as a negative control, while ascorbic acid and quercetin were used as positive controls. The test was repeated three times. Hydroxyl radical scavenging activity was calculated using the equation:

$$\text{Scavenging rate (\%)} = (1 - A_1/A_0) \times 100\%$$

where $A_1$ is the absorbance of the reaction solution containing the sample solution and H2O2, and $A_0$ is the absorbance of the blank solution with distilled water instead of sample solution.

### Superoxide anion radical scavenging assay

The superoxide anion radical scavenging activity assay was conducted using a previous method with modifications (*Vu et al., 2022*). Briefly, methanol solution of fungal crude extract (50 μL, one mg/mL) was added to a 96-well plate and mixed with Tris–HCl buffer solution (100 μL, 50 mmol/L) and pyrogallol solution (18 μL, three mmol/L). Following incubation at room temperature for 3 min, the absorbance of each well was measured at 325 nm. Methanol was used as a negative control, and ascorbic acid and quercetin were used as positive controls. The test was repeated three times. Superoxide anion radical scavenging activity was calculated using the equation:

$$\text{Scavenging rate (\%)} = (1 - A_1/A_0) \times 100\%$$

where $A_1$ is the absorbance of a reaction solution containing sample solution and pyrogallol at 325 nm, and $A_0$ is the absorbance of a blank solution with methanol.

## Data analysis

SPSS software (version 27) was used for statistical processing, and the obtained data was expressed as mean $\pm$ standard deviation. Independent sample Student's $t$-test was used to compare the antibacterial and antioxidant activities between the different fungal crude extracts, and $p < 0.05$ indicates statistical significance.

# RESULTS

## Culture-independent community analysis

A total of 2,130,008 high-quality sequences were obtained by high-throughput analysis of fungal diversity in *A. cerana* gut and honeycomb (Table 1). Among them, 110,412 sequences were obtained from honeybee gut, with an average length of 234.48 bp. The other sequences were obtained from honeycomb, with an average length of 234.53 bp. The rarefaction curves tended to be flat, indicating that the sequencing depth was sufficient to reflect the fungal diversity in the samples (Fig. S1). Alpha diversity analysis was used to estimate the fungal richness and diversity of honeybee gut and honeycomb (Table 1). The Shannon and Simpson diversities of honeybee gut were similar to those of honeycomb, but the Chao1 and ACE diversities of gut were lower than those of honeycomb. The Non-Metric Multi-Dimensional Scaling (NMDS) was used for beta diversity analysis of the fungal communities of ZFCZ and ZFFC (Fig. 1). The results of the NMDS analyses revealed that the fungal communities in the honeybee gut differed significantly from those of the honeycomb.

All sequences were clustered with representative sequences (cutoff 98.65% sequence identity); 652 OTUs were detected (Fig. 2). The number of OTUs found in the honeycomb (506) was greater than that in honeybee gut (280); 134 OTUs were found in both the honeybee gut and honeycomb.

A representative sequence of each fungal OTU was taxonomically classified; the identified fungi belonged to five phyla, 25 classes, 64 orders, 150 families, and 334 genera. At the phylum level, Ascomycota was found to be the most abundant in both the honeybee gut and honeycomb, with relative abundances of 95.57% and 98.20%, respectively (Fig. 3A). This was followed by Basidiomycota with relative abundances of 4.02% and 1.73%, respectively. Mucoromycota, Chytridiomycota, and Olpidiomycota were also observed in small amounts.

At the genus level, 153 and 269 genera were detected in honeybee gut and honeycomb respectively. A difference in the community structure was observed between the honeybee gut and honeycomb (Fig. 3B). *Fusarium* (29.80%), *Stenocarpella* (27.29%), *Botrytis* (13.68%), *Epicoccum* (5.69%), *Aspergillus* (3.17%), *Talaromyces* (2.29%), and *Penicillium* (2.04%) were represented at relatively high abundances in honeybee gut, while *Botrytis* (13.0%), *Periconia* (6.68%), *Aspergillus* (3.81%), *Nigrospora* (3.02%), and *Talaromyces* (1.36%) were the dominant genera in honeycomb.
**Table 1  Statistics of fungal sequencing data and community diversity indices in honeybee samples.**

| Sample | Sequences | Average length (bp) | Observed | Chao1 | ACE | Shannon | Simpson |
|---|---|---|---|---|---|---|---|
| ZFCZ | $36,804 \pm 2,527^a$ | $234.48 \pm 7.44^a$ | $169.67 \pm 16.26^a$ | $192.89 \pm 24.31^a$ | $190.62 \pm 20.84^a$ | $2.45 \pm 0.89^a$ | $0.79 \pm 0.17^a$ |
| ZFFC | $34,198 \pm 3,691^a$ | $234.53 \pm 3.29^a$ | $380.33 \pm 94.34^b$ | $414.47 \pm 89.21^b$ | $419.62 \pm 100.77^b$ | $2.88 \pm 0.86^a$ | $0.80 \pm 0.13^b$ |

**Notes.**
The results are expressed as means $\pm$ standard deviations (SD) from triplicate measurements; the different lowercase letters indicate statistically significant differences in lines ($p < 0.05$).

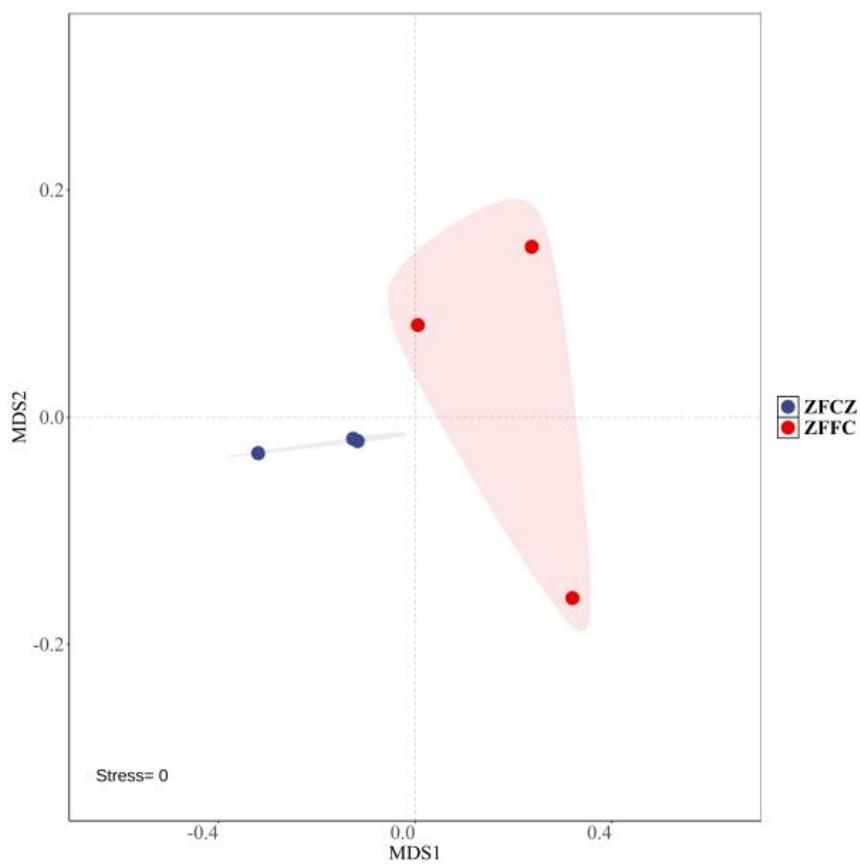

**Figure 1  Non-metric multi-dimensional scaling (NMDS) ordinations based on Bray–Curtis similarities of OTU-based fungal community structures found in honeybee samples.**

## Identification of cultivable fungi

In this study, 28 cultivable fungal strains were isolated. Among them, 17 strains were isolated from honeybee gut, six strains from the head, and five strains from honeycomb (Table 2; Fig. 4). Sequence analysis of the 5.8S rDNA genes showed that these 28 fungal strains could be classified into two phyla, four classes, five orders, eight families, and 10 genera. Twenty-five of the 28 strains belonged to the phylum Ascomycota, including members of three classes (Eurotiomycetes (18 strains), Sordariomycetes (four strains),
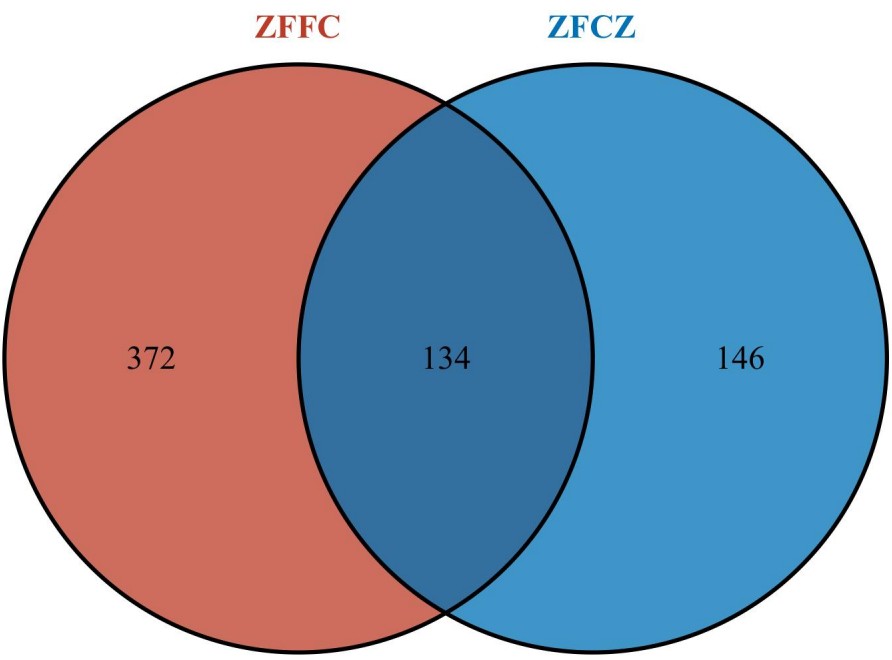

**Figure 2  Fungal OTU Venn diagram of different honeybee samples.**

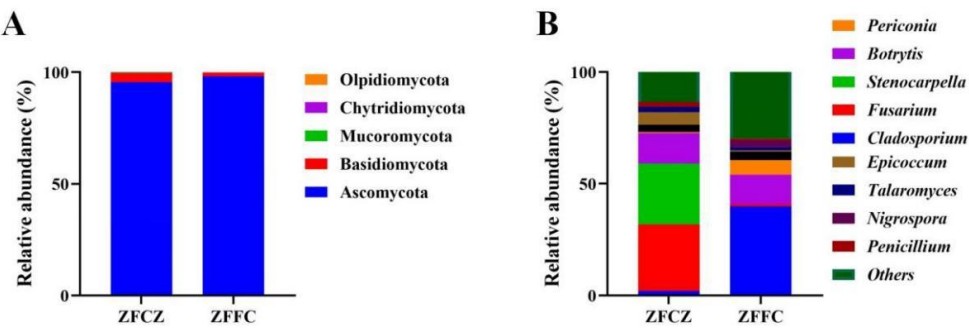

**Figure 3  Analysis of culture-independent microbial communities of honeybee samples.** Relative abundance of OTUs at phylum (A) and genus (B) level. Numbers at nodes are bootstrap scores obtained from 1,000 replications.

and Dothideomycetes (three strains)). The other three strains were members of the class Agaricomycetes in the phylum Basidiomycota.

The largest number of the strains (18/28) was distributed in the class Eurotiomycetes and the order Eurotiales. Of these strains, the most common genera were *Aspergillus* and *Penicillium*. Eleven strains belonging to the genus *Aspergillus* were identified: *A. flavus* (eight strains), *A. aculeatus* (one strain), *A. fijiensis* (one strain), and *A. subramaniani* (one strain). Among them, strains ZFCZ29 and ZFCZ12, isolated from honeybee gut, had 100% sequence similarity to *A. aculeatus* and *A. fijiensis* respectively. Strain ZFFC01 from

**Table 2    The ITS sequence identification of the honeybee-associated fungi.**

| Isolate code | Part | Closest match | Accession number | Coverage/ Max ident | GenBank accession number |
|---|---|---|---|---|---|
| ZFT05 | Head | *Fusarium incarnatum* | MH857681 | 99/100 | PQ898408 |
| ZFT07 | Head | *Cladosporium cladosporioides* | MH865207 | 100/100 | PQ898409 |
| ZFT12 | Head | *Cladosporium cladosporioides* | MH865207 | 100/100 | PQ898410 |
| ZFFC06 | Honeycomb | *Cladosporium tenuissimum* | MH864840 | 100/100 | PQ898411 |
| ZFCZ03 | Gut | *Penicillium adametzioides* | NR103660 | 98/100 | PQ898412 |
| ZFCZ08 | Gut | *Penicillium aeneum* | KP016812 | 100/99.82 | PQ898413 |
| ZFCZ19 | Gut | *Penicillium citrinum* | MH858380 | 100/100 | PQ898414 |
| ZFFC03 | Honeycomb | *Penicillium glabrum* | MH854998 | 100/99.65 | PQ898415 |
| ZFCZ31 | Gut | *Penicillium herquei* | JN626103 | 98/99.64 | PQ898416 |
| ZFCZ15 | Gut | *Penicillium oxalicum* | MH865648 | 100/100 | PQ898417 |
| ZFCZ11 | Gut | *Penicillium sclerotiorum* | MH858515 | 98/99.82 | PQ898418 |
| ZFCZ29 | Gut | *Aspergillus aculeatus* | MH865976 | 100/100 | PQ898419 |
| ZFCZ12 | Gut | *Aspergillus fijiensis* | OL711716 | 100/100 | PQ898420 |
| ZFCZ05 | Gut | *Aspergillus flavus* | OL711682 | 100/99.65 | PQ898421 |
| ZFCZ09 | Gut | *Aspergillus flavus* | OL711682 | 100/99.66 | PQ898422 |
| ZFCZ13 | Gut | *Aspergillus flavus* | OL711682 | 100/99.83 | PQ898423 |
| ZFCZ16 | Gut | *Aspergillus flavus* | OL711682 | 100/99.48 | PQ898424 |
| ZFCZ34 | Gut | *Aspergillus flavus* | MH865138 | 100/100 | PQ898425 |
| ZFFC01 | Honeycomb | *Aspergillus flavus* | OL711682 | 100/100 | PQ898426 |
| ZFT06 | Head | *Aspergillus flavus* | OL711682 | 100/100 | PQ898427 |
| ZFCZ26 | Gut | *Aspergillus flavus* | MH860617 | 100/100 | PQ898428 |
| ZFCZ33 | Gut | *Aspergillus subramanianii* | OL711825 | 100/99.66 | PQ898429 |
| ZFT04 | Head | *Ganoderma lobatum* | JQ520165 | 100/99.66 | PQ898430 |
| *ZFT08 | Head | *Cytospora austromontana* | EU552119 | 98/95.16 | PQ898431 |
| ZFCZ02 | Gut | *Trichoderma citrinoviride* | MH865864 | 98/99.84 | PQ898432 |
| *ZFFC04 | Honeycomb | *Phanerochaete sordida* | MH857075 | 93/96.64 | PQ898433 |
| *ZFFC05 | Honeycomb | *Trametes villosa* | MH856545 | 93/97.17 | PQ898434 |
| ZFCZ24 | Gut | *Aphanocladium album* | MH864426 | 100/99.82 | PQ898435 |

**Notes.**
*potentially novel species.

honeycomb and strain ZFT06 from bee head were both identified as *A. flavus* (sequence identity 100%). Seven strains were identified as belonging to the genus *Penicillium*. Six of these were isolated from honeybee gut and identified as *P. adametzioides*, *P. aeneum*, *P. citrinum*, *P. herquei*, *P. oxalicum*, and *P. sclerotiorum* with >99% sequence identity, respectively. The other *Penicillium* strain, isolated from honeycomb, exhibited a sequence match of 99.65% with *P. glabrum*.

Another observed class was Sordariomycetes (phylum Ascomycota), including members of the orders Hypocreales and Diaporthales. Among them, three strains of Hypocreales were identified as *Fusarium incarnatu*, *Trichoderma citrinoviride*, and *Aphanocladium album*, respectively. Notably, strain ZFCZ02 showed 99.84% sequence similarity to *T. citrinoviride*. Strain ZFT08 isolated from the honeybee head and belonging to the order Diaporthales

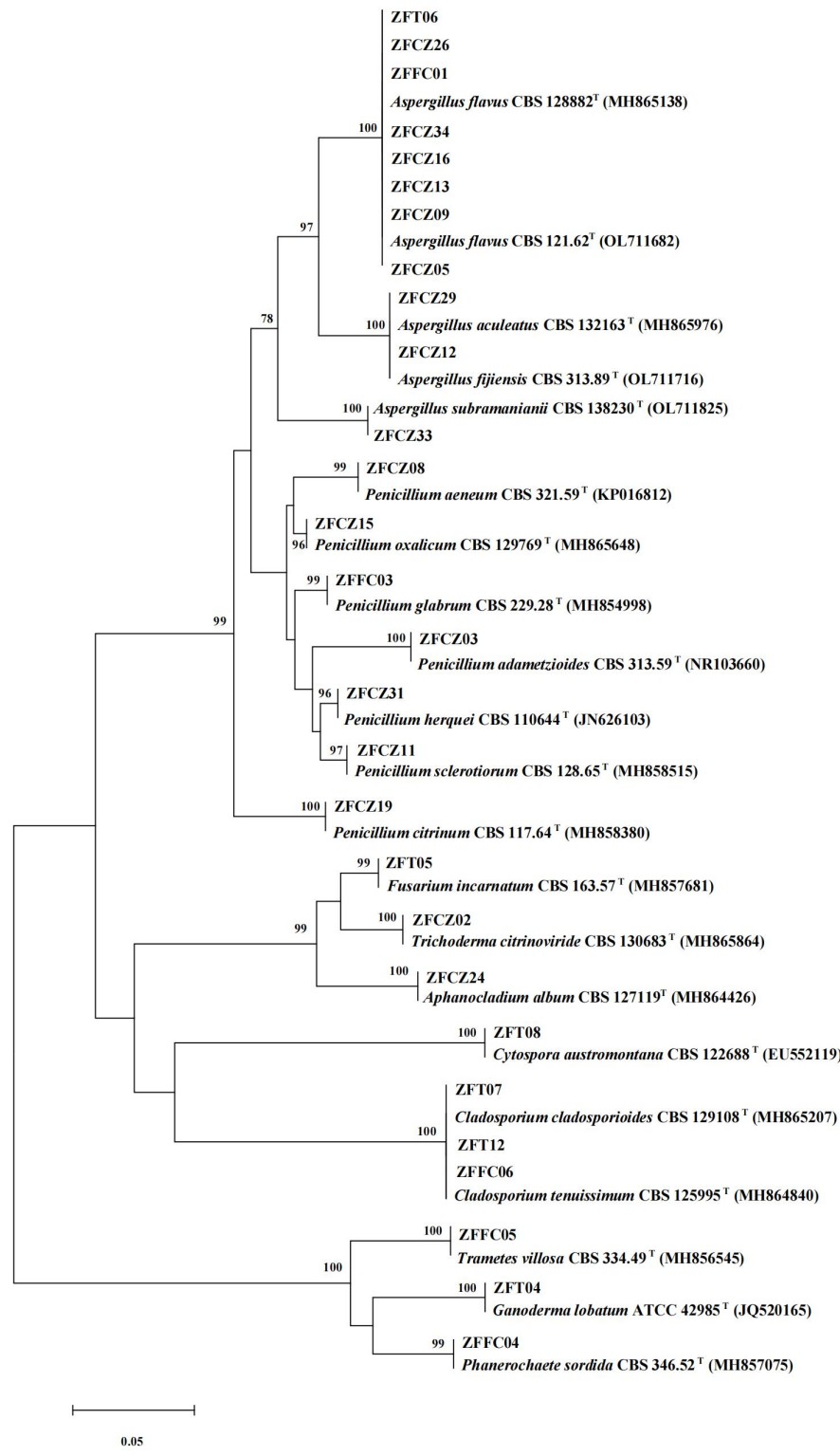

**Figure 4** **Neighbor-joining tree of the ITS sequences of honeybee-associated fungi.** Numbers at nodes are bootstrap scores obtained from 1,000 replications.

showed a best database match similarity of 95.16%, to *Cytospora* aff. *austromontana*; thus, it might be a novel species.

Three Dothideomycetes (phylum Ascomycota) sequences were classified into the order Cladosporiales and the genus *Cladosporium*. Two strains, isolated from honeybee head, were identified as *C. cladosporioides* with sequence identity of 100%. Another strain, ZFFC06, isolated from honeycomb, was identified as *C. tenuissimum* (identity 100%).

Finally, three strains in class Agaricomycetes (phylum Basidiomycota) were grouped into the order Polyporales. Among them, two strains, ZFFC04 and ZFFC05, isolated from honeycomb, showed <98% sequence similarity with *Phanerochaete* aff. *sordida* and *Trametes* aff. *villosa*, respectively, and they may be novel species. The other strain, ZFT04, showed high similarity with Ganoderma lobatum (identity 99.66%).

## Antibacterial activities of cultivable fungi

In this study, crude extracts of fungal secretions in the culture broths of 22 fungal isolates were tested for their antibacterial activities by the filter paper diffusion method (Fig. 5; Table 3). Thirteen of the crude extracts exhibited activity against at least one pathogenic bacterium. Among them, the extract from ZFCZ33 showed the best antibacterial activities against *S. aureus*, *P. aeruginosa*, *E. coli*, and *P. syringae* pv. *Actinidiae* with the disc diameter of inhibition zone diameter (IZD) of 24.33, 15.33, 17.00, and 25.33 mm, respectively, which was signifcantly weaker than the positive gentamicin sulfate ($p < 0.05$). Only ZFCZ33 had antibacterial activities against *P. aeruginosa* and *E. coli*, while other fungal crude extracts did not show antibacterial activity ($p < 0.05$). There are five fungal crude extracts (ZFCZ19, ZFCZ31, ZFCZ12, ZFCZ09, and ZFFC01) that have weak antibacterial activity against *S. aureus,* and their IZD were less than nine mm, significantly lower than the antibacterial activity of ZFCZ33 ($p < 0.05$). Similarly, except for ZFCZ33, nine fungal crude extracts showed antibacterial activity against *P. syringae* pv. *Actinidiae*, among which ZFCZ19 had a moderate antibacterial activity with an IZD of 16.67 mm. Moreover, nine fungal crude extracts exhibited antibacterial activity against *B. subtilis*, with ZFCZ03 having the largest IZD of 10.00 mm and moderate activity compared to the positive control with an IZD of 20.00 mm. Three fungal crude extracts (ZFCZ19, ZFCZ31, and ZFCZ09) showed antibacterial activities against *S. aureus*, *B. subtilis*, and *P. syringae* pv. *actinidiae*. However, none of the tested isolates had antibacterial activity against *B. thuringiensis*.

## Antioxidant activities of cultivable fungi

The antioxidant activities of crude fermentation broth extracts of 22 honeybee-associated fungi were tested (Table 4). The extracts from cultures of ZFT05 and ZFT04 aside, the extracts for the other 20 isolates showed DPPH radical scavenging activity. Among them, extract from ZFCZ03 had the strongest effect on DPPH, with a free radical scavenging rate of 89.71%, which was equivalent to the activities of the positive controls ascorbic acid (92.89%) and quercetin (89.80%). The crude extracts from culture broth of all isolates had radical scavenging activity toward ABTS, and the free radical scavenging rate of 15 isolates toward ABTS was >90%. Strain ZFT07, isolated from honeybee head, had the strongest activity toward ABTS (free radical scavenging rate 97.13%, comparable to the activity of

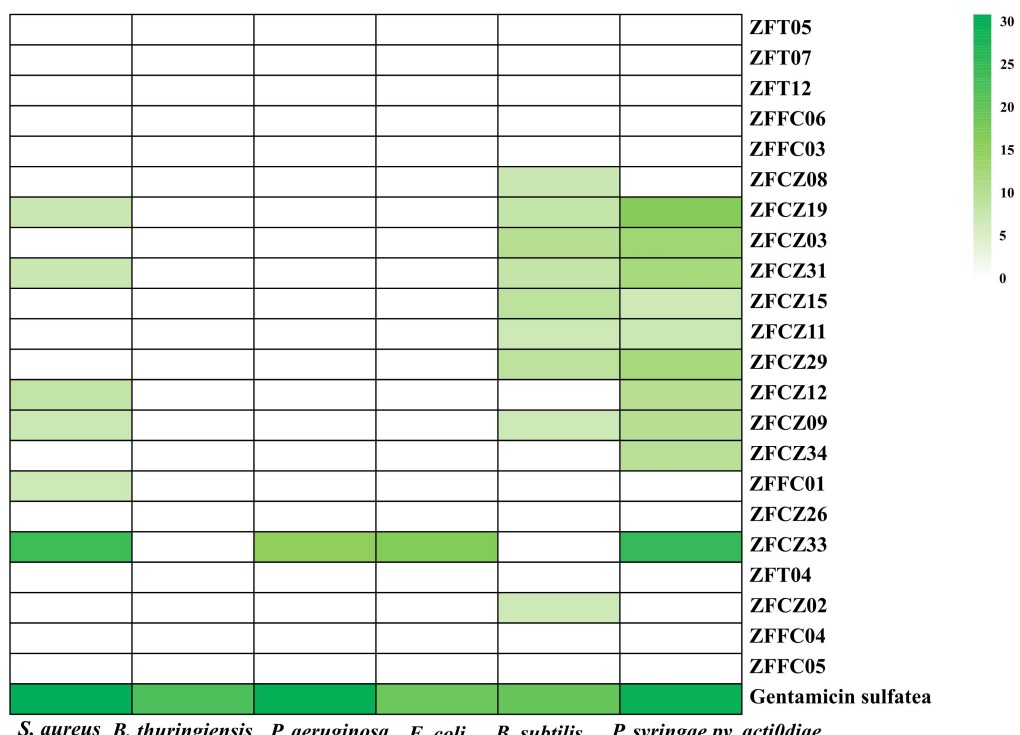

**Figure 5 Heatmap of antibacterial activity crude extracts of honeybee-associated fungi against the tested strains (mm).** Gentamicin sulfate as the positive control of pathogenic bacteria; the concentration for the test is 150 μg/filter paper.

the positive controls ascorbic acid (97.14%) and quercetin (96.95%)). Only weak hydroxyl radical scavenging activity was observed among the fungal extracts. Nineteen fungal culture broth crude extracts showed scavenging capacity for superoxide anion radicals; two isolates showed a scavenging rate of >50%, from strains ZFFC03, ZFCZ19 and ZFCZ03.

## DISCUSSION

Insects are the most numerous animals on earth and coexist with various microorganisms, including bacteria and fungi (*Park et al., 2018*). Insect-associated fungi have attracted increasing attention because of their diversity, which is expected to lead to the development of new resources and antibiotics (*Xu et al., 2020*). Many novel metabolites have been isolated from insect-associated fungi, and some of the metabolites possess antimicrobial and antioxidant activities (*Li et al., 2019*; *Li et al., 2020a*; *Elbanna et al., 2021*). Therefore, here, we used a culture-independent method to analyze the fungal diversity in the gut and honeycomb of the Chinese honeybee *Apis cerana*. Additionally, 28 cultivable fungal strains were isolated from *A. cerana* gut, head, and honeycomb. The antibacterial and antioxidant activities of crude extracts of the fermentation broth of these cultivable fungi were evaluated.

**Table 3  Antibacterial activity of crude extracts of honeybee-associated fungi (inhibition zone diameter: mm).**

| Isolate | S. aureus | B. thuringiensis | P. aeruginosa | E. coli | B. subtilis | P. syringae pv. actinidiae |
|---|---|---|---|---|---|---|
| ZFT05 | 0.00 ± 0.00[f] | 0.00 ± 0.00[b] | 0.00 ± 0.00[c] | 0.00 ± 0.00[c] | 0.00 ± 0.00[g] | 0.00 ± 0.00[h] |
| ZFT07 | 0.00 ± 0.00[f] | 0.00 ± 0.00[b] | 0.00 ± 0.00[c] | 0.00 ± 0.00[c] | 0.00 ± 0.00[g] | 0.00 ± 0.00[h] |
| ZFT12 | 0.00 ± 0.00[f] | 0.00 ± 0.00[b] | 0.00 ± 0.00[c] | 0.00 ± 0.00[c] | 0.00 ± 0.00[g] | 0.00 ± 0.00[h] |
| ZFFC06 | 0.00 ± 0.00[f] | 0.00 ± 0.00[b] | 0.00 ± 0.00[c] | 0.00 ± 0.00[c] | 0.00 ± 0.00[g] | 0.00 ± 0.00[h] |
| ZFFC03 | 0.00 ± 0.00[f] | 0.00 ± 0.00[b] | 0.00 ± 0.00[c] | 0.00 ± 0.00[c] | 0.00 ± 0.00[g] | 0.00 ± 0.00[h] |
| ZFCZ08 | 0.00 ± 0.00[f] | 0.00 ± 0.00[b] | 0.00 ± 0.00[c] | 0.00 ± 0.00[c] | 7.67 ± 0.58[e] | 0.00 ± 0.00[h] |
| ZFCZ19 | 7.67 ± 0.47[d] | 0.00 ± 0.00[b] | 0.00 ± 0.00[c] | 0.00 ± 0.00[c] | 8.33 ± 0.58[d] | 16.67 ± 0.94[c] |
| ZFCZ03 | 0.00 ± 0.00[f] | 0.00 ± 0.00[b] | 0.00 ± 0.00[c] | 0.00 ± 0.00[c] | 10.00 ± 0.82[b] | 13.33 ± 0.94[d] |
| ZFCZ31 | 7.67 ± 0.47[d] | 0.00 ± 0.00[b] | 0.00 ± 0.00[c] | 0.00 ± 0.00[c] | 8.33 ± 0.47[d] | 12.33 ± 0.47[e] |
| ZFCZ15 | 0.00 ± 0.00[f] | 0.00 ± 0.00[b] | 0.00 ± 0.00[c] | 0.00 ± 0.00[c] | 9.00 ± 0.00[c] | 7.00 ± 0.00[g] |
| ZFCZ11 | 0.00 ± 0.00[f] | 0.00 ± 0.00[b] | 0.00 ± 0.00[c] | 0.00 ± 0.00[c] | 7.00 ± 0.00[f] | 7.33 ± 0.47[g] |
| ZFCZ29 | 0.00 ± 0.00[f] | 0.00 ± 0.00[b] | 0.00 ± 0.00[c] | 0.00 ± 0.00[c] | 9.00 ± 0.00[c] | 12.33 ± 0.47[e] |
| ZFCZ12 | 8.67 ± 0.47[c] | 0.00 ± 0.00[b] | 0.00 ± 0.00[c] | 0.00 ± 0.00[c] | 0.00 ± 0.00[g] | 10.00 ± 0.82[f] |
| ZFCZ09 | 7.33 ± 0.47[de] | 0.00 ± 0.00[b] | 0.00 ± 0.00[c] | 0.00 ± 0.00[c] | 7.00 ± 0.00[f] | 10.00 ± 0.00[f] |
| ZFCZ34 | 0.00 ± 0.00[f] | 0.00 ± 0.00[b] | 0.00 ± 0.00[c] | 0.00 ± 0.00[c] | 0.00 ± 0.00[g] | 9.67 ± 0.47[f] |
| ZFFC01 | 7.00 ± 0.00[e] | 0.00 ± 0.00[b] | 0.00 ± 0.00[c] | 0.00 ± 0.00[c] | 0.00 ± 0.00[g] | 0.00 ± 0.00[h] |
| ZFCZ26 | 0.00 ± 0.00[f] | 0.00 ± 0.00[b] | 0.00 ± 0.00[c] | 0.00 ± 0.00[c] | 0.00 ± 0.00[g] | 0.00 ± 0.00[h] |
| ZFCZ33 | 24.33 ± 0.47[b] | 0.00 ± 0.00[b] | 15.33 ± 0.47[b] | 17.00 ± 0.82[b] | 0.00 ± 0.00[g] | 25.33 ± 0.47[b] |
| ZFT04 | 0.00 ± 0.00[f] | 0.00 ± 0.00[b] | 0.00 ± 0.00[c] | 0.00 ± 0.00[c] | 0.00 ± 0.00[g] | 0.00 ± 0.00[h] |
| ZFCZ02 | 0.00 ± 0.00[f] | 0.00 ± 0.00[b] | 0.00 ± 0.00[c] | 0.00 ± 0.00[c] | 7.00 ± 0.00[f] | 0.00 ± 0.00[h] |
| ZFFC04 | 0.00 ± 0.00[f] | 0.00 ± 0.00[b] | 0.00 ± 0.00[c] | 0.00 ± 0.00[c] | 0.00 ± 0.00[g] | 0.00 ± 0.00[h] |
| ZFFC05 | 0.00 ± 0.00[f] | 0.00 ± 0.00[b] | 0.00 ± 0.00[c] | 0.00 ± 0.00[c] | 0.00 ± 0.00[g] | 0.00 ± 0.00[h] |
| Gentamicin sulfate[a] | 31.00 ± 0.82[a] | 22.33 ± 1.25[a] | 30.33 ± 0.47[a] | 19.00 ± 0.82[a] | 20.00 ± 0.82[a] | 29.67 ± 0.47[a] |

**Notes.**
[a]Gentamycin sulfate as the positive control; the results are expressed as means ± standard deviations (SD) from triplicate measurements; the concentration for the test is 150 μ g/filter paper; the different lowercase letters indicate statistically significant differences in lines ($p < 0.05$).

In recent years, high-throughput sequencing technology has been applied widely to reveal the fungal diversity and community structures associated with honeybees (*Callegari et al., 2021*; *Decker et al., 2023*; *Li et al., 2024*). However, compared with the fungal diversity of *A. mellifera* (European honeybee), there are relatively few reports on the fungal diversity of *A. cerana*. Here, we found that Ascomycota was the dominant fungal phylum in both the *A. cerana* gut and honeycomb, similar to previous reports for *A. mellifera* (*Cui et al., 2022a*; *Chow et al., 2024*). However, there were significant differences between the fungal community structures of the gut and honeycomb of *A. cerana* at the genus level. The dominant fungal genera in the gut were *Fusarium*, *Stenocarpella*, *Botryti* s, *Aspergillus*, and *Penicillium*, and some of these may be derived from honeybee food or pollen (*Disayathanoowat et al., 2020*). Meanwhile, *Botrytis*, *Periconia*, *Aspergillus*, *Nigrospora*, and *Talaromyces* were the dominant genera in the honeycomb. The relative abundances of *Botrytis* and *Aspergillus* were similar in the gut and honeycomb. Fungi in the genera *Botrytis* and *Aspergillus* can be pathogens of plants and honeybees, respectively (*Kim et al., 2019*;

**Table 4** Antioxidant activity of crude extracts of honeybee-associated fungi (%).

| Isolate | DPPH radical | ABTS radical | OH radical | $O^{2-}$ radical |
|---|---|---|---|---|
| ZFT05 | $0.00 \pm 0.00^p$ | $10.95 \pm 0.51^g$ | $0.00 \pm 0.00^f$ | $27.45 \pm 7.03^c$ |
| ZFT07 | $56.06 \pm 1.55^{gh}$ | $97.13 \pm 0.04^a$ | $0.00 \pm 0.00^f$ | $17.36 \pm 4.49^{de}$ |
| ZFT12 | $56.56 \pm 0.60^{gh}$ | $97.07 \pm 0.07^a$ | $4.12 \pm 0.41^c$ | − |
| ZFFC06 | $35.77 \pm 0.30^j$ | $30.92 \pm 0.53^f$ | $0.00 \pm 0.00^f$ | $15.46 \pm 2.21^e$ |
| ZFFC03 | $66.39 \pm 0.63^{ed}$ | $96.81 \pm 0.34^a$ | $2.03 \pm 0.10^e$ | $56.72 \pm 2.07^b$ |
| ZFCZ08 | $81.10 \pm 0.94^b$ | $95.76 \pm 0.27^{ab}$ | $1.81 \pm 1.05^e$ | $14.46 \pm 3.20^e$ |
| ZFCZ19 | $69.13 \pm 1.73^{cd}$ | $96.62 \pm 0.41^a$ | $0.00 \pm 0.00^f$ | $59.95 \pm 5.76^b$ |
| ZFCZ03 | $89.71 \pm 0.84^a$ | $96.09 \pm 0.52^a$ | $2.84 \pm 0.81^d$ | $55.30 \pm 10.44^b$ |
| ZFCZ31 | $79.79 \pm 0.61^b$ | $91.79 \pm 0.21^c$ | $0.00 \pm 0.00^f$ | $24.89 \pm 2.29^{cd}$ |
| ZFCZ15 | $63.96 \pm 1.60^{ef}$ | $93.58 \pm 0.25^{bc}$ | $0.00 \pm 0.00^f$ | $10.11 \pm 3.49^e$ |
| ZFCZ11 | $49.97 \pm 1.34^i$ | $93.27 \pm 0.61^{bc}$ | $0.00 \pm 0.00^f$ | $0.00 \pm 0.00^f$ |
| ZFCZ29 | $60.14 \pm 1.67^{fg}$ | $97.18 \pm 0.07^a$ | $3.07 \pm 0.73^d$ | $31.49 \pm 2.62^c$ |
| ZFCZ12 | $72.71 \pm 7.03^c$ | $96.59 \pm 0.21a$ | $0.00 \pm 0.00^f$ | $10.24 \pm 2.81^e$ |
| ZFCZ09 | $55.23 \pm 1.52^h$ | $96.17 \pm 0.39^a$ | $0.00 \pm 0.00^f$ | $18.06 \pm 9.04^{cde}$ |
| ZFCZ34 | $79.55 \pm 6.61^b$ | $96.83 \pm 0.21^a$ | $0.00 \pm 0.00^f$ | $27.34 \pm 3.22^c$ |
| ZFFC01 | $17.20 \pm 2.58^{mn}$ | $92.53 \pm 0.86^c$ | $0.00 \pm 0.00^f$ | $13.16 \pm 3.34^e$ |
| ZFCZ26 | $30.45 \pm 1.38^k$ | $92.65 \pm 0.58^c$ | $0.00 \pm 0.00^f$ | $24.72 \pm 1.38^{cd}$ |
| ZFCZ33 | $25.90 \pm 0.48^l$ | $83.08 \pm 3.60^d$ | $0.00 \pm 0.00^f$ | $24.33 \pm 1.80^{cd}$ |
| ZFT04 | $0.00 \pm 0.00^p$ | $4.70 \pm 3.34^h$ | $0.00 \pm 0.00^f$ | $14.67 \pm 0.81^e$ |
| ZFCZ02 | $20.75 \pm 0.70^m$ | $74.97 \pm 0.97^e$ | $0.00 \pm 0.00^f$ | $13.17 \pm 1.33^e$ |
| ZFFC04 | $13.19 \pm 0.09^{no}$ | $12.34 \pm 1.70^g$ | $0.00 \pm 0.00^f$ | $25.53 \pm 0.53^{cd}$ |
| ZFFC05 | $11.34 \pm 0.08^o$ | $11.07 \pm 1.22^g$ | $0.00 \pm 0.00^f$ | $0.00 \pm 0.00^f$ |
| Ascorbic acid[a] | $92.89 \pm 0.30^a$ | $97.14 \pm 0.27^a$ | $93.45 \pm 0.16^a$ | $94.21 \pm 0.95^a$ |
| Quercetin[a] | $89.80 \pm 0.42^a$ | $96.95 \pm 0.30^a$ | $15.06 \pm 0.43^b$ | $97.94 \pm 1.46^a$ |

Notes.
[a] Ascorbic acid and quercetin as the positive control; the results are expressed as means ± standard deviations (SD) from triplicate measurements; the different lowercase letters indicate statistically significant differences in lines ($p < 0.05$).

*Becchimanzi & Nicoletti, 2022*). Honeybees can not only pollinate fruits but also control *Botrytis* that causes plant gray mold disease (*Kapongo et al., 2008*).

Using the culture-independent method, 153 and 269 fungal genera were detected from the honeybee gut and honeycomb, respectively. Using the culture-dependent method, four and five genera were isolated, respectively. Culture-independent methods typically show higher fungal composition and diversity than culture-dependent methods; this provides motivation for the development of strategies to obtain more fungi from honeybee samples in future studies, for example by improving culture methods and conditions, and media-fungus pairings (*Oberhardt et al., 2015*; *Li et al., 2023*). The main species isolated from the honeybee gut, head, and honeycomb were *Aspergillus*, *Penicillium* and *Cladosporium* spp., *Aspergillus*, *Cladosporium*, *Mucor*, *Penicillium*, *Rhizopus*, and *Talaromyces* have been reported to inhibit the growth of the entomopathogen *Ascosphaera apis* (*Rutkowski, Weston & Vannette, 2023*). Furthermore, these fungal genera have been isolated from the guts of other insects, including termites, dragonflies, and beetles (*Moubasher, Abdel-Sater & Zeinab, 2017*; *Xu et al., 2020*; *Cui et al., 2022b*). *A. flavus*, isolated here from the gut of

*A. cerana*, has previously been found to be a honeybee pathogen that caused stonebrood (*Foley et al., 2014*; *Miller, Smith & Newton, 2021*). Moreover, some plant pathogenic fungi were also isolated from the honeybee samples in the present work, such as *F. incarnatum* and *C. cladosporioides* (*Nam et al., 2015*; *Ekabote et al., 2023*). Several strains isolated here are reported as being associated with *A. cerana* for the first time, for instance *Ganoderma lobatum*, and *Aphanocladium album*.

Most of the crude extracts of the fermentation broths of honeybee-associated fungi in our study did not exhibit strong antibacterial activity, consistent with a previous report (*Cui et al., 2022a*). Similarly, it was found that some crude extracts of termite- and dragonfly-associated fungi did not exhibit antibacterial activity (*Xu et al., 2020*; *Kong et al., 2025*). The fungal strains with antibacterial activity were mainly isolated from the *A. cerana* gut, especially isolates belonging to the genera *Aspergillus* and *Penicillium*. Some insect-associated *Aspergillus* and *Penicillium* spp. have been reported to have antibacterial activities and have also become important sources of novel compounds (*Xu et al., 2020*; *Wang et al., 2021*; *Nayak, Prabhakar & Nanda, 2022*). In addition, it has been reported that there is a dynamic symbiotic relationship between *Aspergillus* and honeybees, ranging from mutualism to antagonism (*Becchimanzi & Nicoletti, 2022*). Specifically, *Aspergillus* spp. produce inhibitory effects towards bee pathogens (*Vojvodic et al., 2011*). Notably, *A. subramanianii* strain ZFCZ33 was a rare fungus of the *Aspergillus* genus from the *A. cerana* gut, and its crude extract from the culture medium displayed strong antibacterial activity against *P. syringae* pv. *Actinidiae.* Therefore, this fungus can be further considered for exploring its antibacterial compounds for the treatment of kiwifruit bacterial canker, which is caused by *P. syringae* pv. *actinidiae* (*Huang et al., 2023*).

Many studies have shown that insect-associated fungi have good antioxidant activity and they have become an important source of novel active antioxidant compounds (*Cui et al., 2022a*; *Elbanna et al., 2021*; *Li et al., 2019*; *Li et al., 2020a*). In this study, the crude extracts from cultures of *A. cerana*-associated fungi showed varying levels of antioxidant activity. Extract from culture of *P. adametzioides* ZFCZ03 showed the strongest antioxidant activity in reduction of DPPH radicals, and that from *C. cladosporioides* ZFT07 showed the strongest antioxidant activity in assay using ABTS. Previously, *C. cladosporioides* OP870014 isolated from *Cordia dichotoma* was reported to have strong antimicrobial, antioxidant, and anticancer activities (*Sharma et al., 2023*). Most isolates of *Aspergillus* have strong antioxidant activity against ABTS radicals. In addition, insect-associated *Aspergillus* has become an important source of novel antioxidant compounds (*Li et al., 2016*; *Li et al., 2020b*; *Shan et al., 2020*). Many previous studies have shown that polyphenolic substances in fungal crude extracts play an important role in antioxidant activity (*Rocha et al., 2020*; *Tang et al., 2021*; *Ghazi-Yaker et al., 2024*). Here, ethyl acetate was used as the extraction solvent to selectively extract low-molecular-weight phenolic compounds and high-molecular-weight polyphenols (*Gurgel et al., 2023*). Our preliminary results revealed that honeybee-associated fungi exhibited more excellent antibacterial activity against Gram-positive bacteria than Gram-negative bacteria. The active metabolites from the fungal strains isolated in the present work should be explored further.

## CONCLUSIONS

This study preliminarily elaborates the fungal diversity of *A. cerana* by using both culture-independent and -dependent methods and expands knowledge about the fungal species in the bee gut and honeycomb. Crude extracts of culture of *A. subramanianii* ZFCZ33 had strong antibacterial activity; this strain may have the potential for further investigation as a bioactive agents. Crude extracts of cultures of *P. adametzioides* ZFCZ03 and *C. cladosporioides* ZFT07 showed potent antioxidant activities against DPPH and ABTS radicals, respectively, and they may thus have the potential for further investigation as a bioactive agents. Together, this study suggests that honeybee-associated fungi are a promising source of biologically active secondary metabolites.

## ACKNOWLEDGEMENTS

We thank Liwen Bianji at Edanz for editing the language of a draft of this manuscript.

### Funding

This work was supported by the Natural Science Key Project of Anhui Province's Education Department (2024AH051755), the Program for Excellent Scitech Innovation Teams of Universities in Anhui Province (2023AH010054), the Talent Program of Huangshan University (2023xkjq007), the University Synergy Innovation Program of Anhui Province (GXXT-2023-054), the Huizhou Mushroom Industry and Microbial Technology Innovation Center of Huangshan University (kypt202001), and First-class Discipline of Huangshan University (ylxk202101). The funders had no role in study design, data collection and analysis, decision to publish, or preparation of the manuscript.

### Grant Disclosures

The following grant information was disclosed by the authors:
Natural Science Key Project of Anhui Province's Education Department: 2024AH051755.
Program for Excellent Scitech Innovation Teams of Universities in Anhui Province: 2023AH010054.
Talent Program of Huangshan University: 2023xkjq007.
University Synergy Innovation Program of Anhui Province: GXXT-2023-054.
Talent Program of Huangshan University: 2023xkjq007.
University Synergy Innovation Program of Anhui Province: GXXT-2023-054.
Huizhou Mushroom Industry and Microbial Technology Innovation Center of Huangshan University: kypt202001.
First-class Discipline of Huangshan University: ylxk202101.

### Competing Interests

The authors declare there are no competing interests.

## Author Contributions

- Pu Cui performed the experiments, analyzed the data, prepared figures and/or tables, authored or reviewed drafts of the article, and approved the final draft.
- Guanxiu Guan performed the experiments, analyzed the data, prepared figures and/or tables, and approved the final draft.
- Zhuoting Gan analyzed the data, authored or reviewed drafts of the article, and approved the final draft.
- Ting Yao conceived and designed the experiments, authored or reviewed drafts of the article, and approved the final draft.

## Data Availability

The ITS gene sequences are available at GenBank: PQ898408–PQ898435, PRJNA1213968.

## Supplemental Information

Supplemental information for this article can be found online at http://dx.doi.org/10.7717/peerj.19762#supplemental-information.

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
