# Peer review of "Diversity, antibacterial and antioxidant activities of fungi associated with Apis cerana"

_PeerJ, doi:10.7717/peerj.19762_

## Round 0.1 · original submission · Major Revisions

The three reviewers have taken the time to provide extensive feedback please address all their comments for further consideration.

·

Basic reporting

Overall, the manuscript is well-written and structured. However, slight grammatical improvements are needed to enhance clarity and readability.

Experimental design

I recommend adding statistical analysis for both biological assays to ensure the results are statistically validated and reproducible.

Validity of the findings

Some tables were not mentioned in the results section. I recommend referencing all tables appropriately to ensure clarity and coherence in data presentation

Additional comments

The authors report on the “Antibacterial activity of the endophytic fungal extracts and synergistic effects of combinations of EDTA against Pseudomonas aeruginosa and Escherichia coli”. The authors address the following reviewer comments and suggestions:

1. Include relevant citations in the methodology section, as it follows standard methods. This will enhance the credibility and traceability of the techniques used.
2. The results section should provide a descriptive account of findings rather than simply referring to figures and tables. Summarize key findings in words, connecting them to specific figures or tables, to give readers a clear interpretation.
3. Ensure data are presented as mean ± standard deviation (SD) in tables, as mentioned in the text. This consistency will strengthen the scientific rigor of the data presentation.
4. Clarify the statistical significance of results within the results section, describing how significance was calculated and providing p-values where relevant.
5. Improve the interpretation of results by explaining the biological or practical implications, helping readers understand the importance of the findings.
6. Consider comparing antibacterial activity against multiple positive controls, taking the mode of action into account to enrich the discussion and interpretation of assay results.
7. Focus on narrating the data in an engaging way to highlight its relevance and make the results more compelling for readers.
8. In lines 357-358, there’s a need for clarity regarding “Our preliminary results revealed that endophytic fungi exhibited more excellent antibacterial activity against Gram-positive bacteria than Gram-negative bacteria”. Specify the origin of this statement to avoid confusion.
9. Review the entire manuscript to ensure that citations follow the journal’s guidelines, maintaining consistency in referencing style.

·

Basic reporting

This research is clearly written in professional, unambiguous English and provides sufficient background and context through appropriate literature references. The article follows a well-organized structure with relevant figures, tables, and shared raw data, making it a self-contained study that effectively presents results aligned with its hypotheses. However, some attention is needed to ensure that the in-text citations and reference list fully adhere to the journal’s formatting guidelines. Addressing this will enhance the overall professionalism and readability of the manuscript.

Experimental design

This study represents original primary research that aligns well with the aims and scope of the journal. The research question is clearly defined, relevant, and addresses a meaningful gap in current knowledge regarding the fungal diversity and bioactivity associated with Apis cerana. The investigation was conducted rigorously, adhering to high technical and ethical standards. Additionally, the methods are described in sufficient detail to ensure reproducibility, supporting the study’s credibility and potential for further scientific advancement.

Validity of the findings

A key strength of this research is that it reveals the rich and previously underexplored fungal diversity associated with Apis cerana, identifying not only a wide range of fungal taxa but also demonstrating their potential as sources of bioactive compounds. By using both culture-dependent and independent methods, the study provides a comprehensive overview of fungal communities in the honeybee gut and honeycomb. The discovery of strains with strong antibacterial and antioxidant activities highlights their potential applications in medicine, agriculture, and biotechnology, laying the groundwork for future exploration and practical use of honeybee-associated fungi.

Additional comments

This study highlights the untapped fungal diversity in Apis cerana and reveals strains with strong antibacterial and antioxidant properties. Combining culture-based and sequencing methods offers new insight into honeybee-associated fungi and their potential for biotechnological applications. This manuscript provided good knowledge and had the potential to be accepted, but some important points have to be clarified or fixed.

Overall, this study highlights the untapped fungal diversity in Apis cerana and reveals strains with strong antibacterial and antioxidant properties. Combining culture-based and sequencing methods offers new insight into honeybee-associated fungi and their potential for biotechnological applications. This manuscript provided good knowledge and had the potential to be accepted, but some important points have to be clarified or fixed.

1. What is the scientific significance of this study?
2. Please review and adjust the in-text citations and reference list to ensure they conform to the journal’s formatting guidelines
Introduction
3. This introduction works well because it highlights the importance of honeybee gut fungi, a less-studied but valuable area. It shows their potential roles in bee health and bioactive compound production and clearly explains the need to study fungal diversity in Apis cerana, making the research timely and relevant.
Materials & Methods
4. Lines 72-78: Please provide more details, how many honey bee colonies were sampled
5. Lines 102-111: Please provide more details, what technique do you use to run the experiment (pour plate or spread plate), and add more details about this.
6. Lines 102-111: Please add the instrument (incubator with the brand and serial number) to the method of incubation fungi
7. Lines 102-111: How long do you incubate the inoculated plate
8. Lines 136-137: Please provide the full name of LBA medium
Results and Discussion
9. The authors provide good data that is easy to understand. And the discussion is very good and clear

References
10. Line 42: Please replace “Raymann et al., 2018” with “Raymann & Moran, 2018”
11. Line 42: Please replace “Motta et al., 2024” with “Motta & Moran, 2024”
12. Line 46: Please replace “Rutkowski et al., 2023” with “Rutkowski, Weston & Vannette, 2023”
13. Line 47: Please replace “Hsu et al., 2021” with “Hsu, Wang & Wu, 2021”
14. Line 47: Please replace “Tiusanen et al., 2024” with “Tiusanen, Becker-Scarpitta & Wirta, 2024”
15. Line 53: Please replace “Wynns et al., 2013” with “Wynns, Jensen & Eilenberg, 2013”
16. Lines 60-61: Please replace “Elbanna et al., 2021a” with “Elbanna et al., 2021”
17. Line 61: Please replace “Elbanna et al., 2021b” with “Elbanna, Khalil & Capon, 2021”
18. Line 294: Please check “Li et al., 2020” to relate with reference list, and please use only one of these “Li et al., 2020a” or “Li et al., 2020b”
19. Line 294: Please check “Elbanna et al., 2022a”; it is not in the reference list. Please remove, add, or make it correct.
20. Line 312: Please replace “Becchimanzi et al., 2022” with “Becchimanzi & Nicoletti, 2022”
21. Line 324: Please replace “Rutkowski et al., 2023” with “Rutkowski, Weston & Vannette, 2023”
22. Line 326: Please replace “Moubasher et al., 2017” with “Moubasher, Abdel-Sater & Zeinab, 2017”
23. Line 328: Please replace “Miller et al., 2021” with “Miller, Smith & Newton, 2021”
24. Moubasher et al., 2017
25. Line 339: Please replace “Nayak et al., 2022” with “Nayak, Prabhakar & Nanda, 2022”
26. Line 346: Please check “Li et al., 2020” to relate with reference list, and please use only one of these “Li et al., 2020a” or “Li et al., 2020b”
27. Line 354: Please check “Li et al., 2020” to relate with reference list, and please use only one of these “Li et al., 2020a” or “Li et al., 2020b”
28. Line 443: Please replace “2021a” with “2021”
29. Line 446: Please replace “2021b” with “2021”
30. Line 489: Please replace “2020” with “2020a”
31. Line 492: Please replace “2020” with “2020b”

Reviewer 3 ·

Basic reporting

The study addresses a relevant topic by combining fungal diversity profiling with bioactivity screening. However, the manuscript requires substantial revision before it can be considered for publication. The main concerns relate to the lack of statistical analysis, insufficient methodological detail, limited integration of figures and tables, and overinterpretation of preliminary findings. My observations are listed below:

The knowledge gap is not clearly defined. Please clearly state why studying fungal diversity in Apis cerana is important, and how this study provides new insight compared to existing work on A. mellifera or other bees. Indicate what is unknown and how your study addresses it.

There are no statistical comparisons between gut and honeycomb fungal communities. The authors should perform statistical analysis of alpha diversity metrics (e.g., Shannon index) using t-tests or non-parametric alternatives. Consider using PERMANOVA or NMDS for beta diversity comparisons.

No standard deviations, p-values, or significance tests reported for antibacterial and antioxidant results. Please document how many replicates were performed per fungal extract, provide standard deviation/error bars for inhibition zones and antioxidant activities, and conduct and report statistical tests to determine if differences are significant (e.g., one-way ANOVA with post-hoc Tukey).

Rephrase conclusions to reflect that findings are preliminary and require compound-level validation. Use cautious language, e.g., “may have potential for further investigation as bioactive agents” instead of “could serve as antibiotics.”

Please confirm whether the sequence identity was ≥99%. Indicate which isolates were identified at the species level with confidence. If identification is uncertain, label them as “cf.” or “aff.” accordingly.
All figures presenting quantitative data (e.g., antioxidant and antibacterial activities) must include error bars (standard deviation or standard error) and statistical significance indicators where appropriate. Figure captions should clearly state the number of replicates and the units used. Without this information, the reliability and comparability of the results cannot be properly evaluated.

Experimental design

The experimental design lacks sufficient replication and statistical analysis. Please clarify the number of biological and technical replicates for each assay, apply appropriate statistical tests to the results, and include measures of variability (e.g., standard deviation). Additionally, justify the limited sample size and address its impact on the result interpretation.

Validity of the findings

The findings are preliminary and largely descriptive. Conclusions about bioactivity and potential applications are overstated and not supported by compound-level data. Please moderate the claims and clearly state that further validation is required.

---

## Round 0.2 · accepted · Accept

Thank you for responding to the reviewers' feedback. I am happy to accept the manuscript for publication.

·

Basic reporting

-

Experimental design

-

Validity of the findings

-